

# Thermalization of long range Ising model in different dynamical regimes: A full counting statistics approach

Nishan Ranabhat[1,2] and Mario Collura[1]

**1** SISSA - International School for Advanced Studies, via Bonomea 265, 34136 Trieste, Italy
**2** The Abdus Salam International Centre for Theoretical Physics,
Strada Costiera 11, 34151 Trieste, Italy

⋆ nranabha@sissa.it

## Abstract

We study the thermalization of the transverse field Ising chain with a power law decaying interaction $\sim 1/r^\alpha$ following a global quantum quench of the transverse field in two different dynamical regimes. The thermalization behavior is quantified by comparing the full probability distribution function (PDF) of the evolving states with the corresponding thermal state given by the canonical Gibbs ensemble (CGE). To this end, we used the matrix product state (MPS)-based Time Dependent Variational Principle (TDVP) algorithm to simulate both real time evolution following a global quantum quench and the finite temperature density operator. We observe that thermalization is strongly suppressed in the region with strong confinement for all interaction strengths $\alpha$, whereas thermalization occurs in the region with weak confinement.

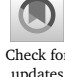

## 1   Introduction

The investigation of non-equilibrium dynamics in isolated many-body systems has garnered significant attention in recent decades, owing to advancements in the manipulation of synthetic quantum systems in laboratory settings [1–11] and the development of analytical and numerical techniques [12–25]. An enduring question in quantum many-body dynamics pertains to the potential thermalization of a closed system that has been perturbed from equilibrium. Thermalization implies that the long-term behavior of a dynamical system can be anticipated using the principles of statistical mechanics. Generally, in the case of a non-integrable closed system, one would expect thermalization in accordance with the Eigenstate Thermalization Hypothesis (ETH) [26–30]. Nonetheless, certain studies have presented contradictory evidence, at least within their specific regime and time scales of investigation [31–34]. In a scenario where a closed system is initially prepared in a generic state, denoted as $|\psi_i\rangle$ (which is not an eigenstate of the Hamiltonian), and subsequently evolved unitarily with a non-integrable Hamiltonian, $\hat{H}$, thermalization is said to occur if the local observables eventually relax to an equilibrium state that corresponds to the predictions of the thermal ensemble,

$$\langle\hat{O}\rangle_{t|t\to\infty} \to \langle\hat{O}\rangle_{\text{eq}} = \langle\hat{O}\rangle_{\text{MCE}} = \frac{1}{N_{E_i,\delta\mathcal{E}}} \sum_{|E_n-E_i|<\delta\mathcal{E}} \langle\psi_n|\hat{O}|\psi_n\rangle \,, \tag{1}$$

$$\langle\hat{O}\rangle_{t|t\to\infty} \to \langle\hat{O}\rangle_{\text{eq}} = \langle\hat{O}\rangle_{\text{CGE}} = \frac{\text{Tr}(\hat{\rho}_\beta\hat{O})}{\text{Tr}(\hat{\rho}_\beta)} \,. \tag{2}$$

Equation (1) indicates thermalization according to the microcanonical ensemble (MCE). The summation encompasses all the eigenstates of the Hamiltonian within a narrow energy range $\delta\mathcal{E}$ centered around the initial energy $E_i = \langle\psi_i|\hat{H}|\psi_i\rangle$. The normalization factor $N_{E_i,\delta\mathcal{E}}$ tallies the energy eigenstates within this range, spanning $2\delta\mathcal{E}$. This approach requires either full diagonalization [28, 35, 36] or partial diagonalization centered on the initial energy density [37] of the Hamiltonian. Consequently, computational limitations arise as a function of the system size. Equation (2) implies thermalization in accordance with the canonical Gibbs ensemble (CGE). The trace is performed over the density operator $\hat{\rho}_\beta$, defined as the inverse temperature $\beta = 1/T$, which is determined by the system's initial energy $E_i = \frac{\text{Tr}(\hat{\rho}_\beta\hat{H})}{\text{Tr}(\hat{\rho}_\beta)}$. Further elaboration on how to extract $\beta$ and $\hat{\rho}_\beta$ is provided in section 3.2. Equations (1) and (2) are recognized as the conditions for strong thermalization. An alternative weak thermalization condition occurs when the time-averaged local order parameter converges to the thermal prediction [33, 34].

Disordered systems demonstrating many-body localization impede thermalization [38–41]. In clean systems, dynamical confinement hinders information propagation and the thermalization process [42–53]. The Long Range Ising model (LRIM) exhibits confinement [44,45] as outlined in Appendix D. A recent study [4] observed the suppression of thermalization in the confined regime of LRIM simulated with trapped ions. Furthermore, employing high-scale exact diagonalization, the validity of ETH has been examined for various interaction strength parameters, denoted as $\alpha$ (see section 2), in LRIM. The results indicate that a strong ETH typically holds at least within the range $\alpha \geq 0.6$ [35]. In our previous work [54], we explored the relaxation of order parameter statistics following a global quench, revealing two distinct dynamical regimes based on the gaussification of the full counting statistics (FCS) of subsystem magnetization. Building on this foundation, the present study investigates the thermalization of LRIM under the CGE framework following a global quench into different dynamical regimes. We evaluate thermalization using the most rigorous criteria by comparing the FCS of the time-evolving state post-global quench with that of the corresponding thermal state.

The remainder of this paper is organized as follows: In Section 2, we introduce the model, the order parameter, its distribution, and a metric for quantifying the proximity of the time-evolved state to the thermal state. Section 3 provides comprehensive details of the numerical methods, quench protocol, and extraction of effective temperatures associated with the global quench. Specific details on simulating finite temperature density operators and calculating the full counting statistics of the order parameter is provided in Appendix B.1 and Appendix B.2 respectively. Section 4 presents the outcomes of our study. Appendix B.3 details the error analysis of the numerical results and Appendices C and D provide supplementary information on thermal phase transitions and correlation propagation in the Long Range Ising model. Finally, we conclude by summarizing our findings and suggesting potential avenues for future research in Section 5.

## 2 Model and Methods

We investigate the ferromagnetic long-range Ising model (LRIM) described by the Hamiltonian in Equation (3),

$$\hat{H}(J,\alpha,h) = -\frac{1}{\mathcal{K}(\alpha)} \sum_{i<j}^{N} \frac{|J|}{|i-j|^\alpha} \hat{s}_i^x \hat{s}_j^x - h \sum_{i=1}^{N} \hat{s}_i^z, \tag{3}$$

where $\hat{s}_i^\mu, \mu = x, y, z$ is the spin one-half matrices at site $i$. We consider open boundary condition, that is relevant to existing experimental setups. The ferromagnetic interaction between two spins falls as the inverse power of the distance between them and is parameterized by the interaction strength parameter $\alpha$. For $\alpha \leq 1$, the inverse power-law interaction series diverges with the lattice size and is normalized using the Kac normalization constant, as defined in Equation (4).

$$\mathcal{K}(\alpha) = \frac{1}{N-1} \sum_{i<j}^{N} \frac{1}{|i-j|^\alpha} = \frac{1}{N-1} \sum_{n=1}^{N} \frac{N-n}{n^\alpha}. \tag{4}$$

This normalization ensures the intensivity of the energy density in the regime $\alpha \leq 1$. The static and dynamic behaviors of this model are strongly influenced by the interaction strength parameter $\alpha$. At $\alpha = \infty$, the model simplifies to the transverse field Ising model (TFIM), which can be solved exactly by mapping it to a system of spinless fermions through Jordan-Wigner transformations [55]. TFIM exhibits a quantum phase transition from the ferromagnetic phase

to the paramagnetic phase at $h = |J|/2$. This quantum phase transition persists as $\alpha$ decreases, with the transition point shifting towards higher values of the magnetic field $h$ [5, 56, 57]. At the opposite extreme of $\alpha = 0$, we have a fully connected regime that is amenable to analytical treatment for both the static and dynamic properties [54, 58, 59]. For $\alpha < 2$, this model displays a long-range ferromagnetic order at low finite temperatures [60, 61]. Given the absence of spontaneous symmetry breaking in finite systems and the $\mathbb{Z}_2$ symmetry of the model in Equation (3), the finite-temperature states with ferromagnetic order also exhibit $\mathbb{Z}_2$ symmetry (see appendix C). The regime with $\alpha < 2$ is particularly intriguing and features a wealth of exotic phenomena such as prethermalization [11], nonlinear propagation of light cones [62, 63], dynamical phase transitions [2, 3, 54, 64–69], and dynamical confinement [4, 44, 45]. Furthermore, this model has garnered significant attention owing to its experimental relevance, particularly in systems involving trapped ions with adjustable transverse field strengths and interaction ranges [1–4, 11].

The complete information of a generic time evolving quantum state, expanded in the computational basis $|\psi_t\rangle = \sum_{\{\sigma_i\}} C_{\{\sigma_i\}}(t)|\sigma_1, \sigma_2, \ldots, \sigma_i, \ldots, \sigma_N\rangle$, is encapsulated within the set of time dependent coefficients $\{C_{\{\sigma_i\}}(t)\}$. In many-body systems, these coefficients scales exponentially with the number of spins, rendering their study exceedingly challenging. A common approach for investigating dynamics in such systems is to monitor the evolution of the expectation value of a local observable $\langle\psi_t|\hat{O}|\psi_t\rangle$, such as the order parameter in systems exhibiting order-disorder transitions. A more robust strategy involves tracking the full probability distribution function (PDF) of this observable, which provides comprehensive information on quantum fluctuations in the system, including all moments and cumulants. Specifically, when the operator $\hat{O}$ is diagonal in computational basis, the corresponding PDF is defined as

$$P(O, t) = \sum_{\{\sigma_i\}} \left|C_{\{\sigma_i\}}(t)\right|^2 \, \delta\left(O - \langle\{\sigma_i\}|\hat{O}|\{\sigma_i\}\rangle\right) \tag{5}$$

which represents the histogram of the squared coefficients of the many-body wave function within the range of possible outcomes of the measurements of $\hat{O}$. The PDF allows for straightforward calculation of moments (or cumulants) of any order. In this study, we employ the eigenvectors of the total spin operator in the longitudinal direction, that is, $\hat{S}^x = \sum_{i=1}^{N} \hat{s}_i^x$, as our computational basis. In this context, the order parameter of interest is the longitudinal magnetization defined for a subsystem of size $l$ within a system of $N$ spins:

$$\hat{M}_l = \sum_{i=1}^{l} \hat{s}_i^x . \tag{6}$$

The observable $\hat{M}_l$ is suitable because it typically relaxes to a stationary state [70], ultimately approaching a stationary statistical distribution in a subsystem of dimension $l$. In addition, because $\hat{M}_l$ is diagonal in computational basis, the definition in Equation (5) applies. The probability distribution function of the subsystem magnetization $\hat{M}_l$ within a generic state $\hat{\rho}_t$ (whether pure or mixed) is given by

$$P_l(m, t) = \text{Tr}(\hat{\rho}_t \delta(\hat{M}_l - m)), \tag{7}$$

which can be Fourier-transformed into an integral form:

$$P_l(m, t) = \int_{-\pi}^{\pi} \frac{d\theta}{2\pi} e^{-i\theta m} \text{Tr}\left(\hat{\rho}_t e^{i\theta\hat{M}_l}\right), \tag{8}$$

where $G_l(\theta, t) = \text{Tr}\left(\hat{\rho}_t e^{i\theta\hat{M}_l}\right)$ denotes the moment-generating function. Given that the Hamiltonian (3) involves a system of spin one-half particles, the values of $m$ span either integers

or half-integers within the range $m \in \left\{ -\frac{l}{2}, -\frac{l}{2}+1, \ldots, \frac{l}{2}-1, \frac{l}{2} \right\}$ depending on whether $l$ is even or odd. In Appendix B.2 we illustrate the detailed calculation of $G_l(\theta, t)$ with matrix product state (MPS) representation. Historically, the PDF has been studied as the full counting statistic (FCS) of electron fluctuations in mesoscopic systems [71–73]. More recently, FCS has been explored in quantum many-body systems in both equilibrium and non-equilibrium scenarios [18, 19, 43, 54, 74–79].

To assess thermalization, we introduce a metric called "Distance to Thermalization", DT($t$), initially introduced in [80]. This metric quantifies the Euclidean distance between the probability distribution function (PDF) of the order parameter at time $t$ following a quantum quench, denoted as $P_l(m, t)$, and the corresponding thermal PDF, represented as $P_l^{\text{TH}}(m)$. Mathematically, it is defined as

$$\text{DT}(t) = \sqrt{\sum_m \left[ P_l(m, t) - P_l^{\text{TH}}(m) \right]^2}. \tag{9}$$

It is noteworthy that the convergence of the PDF provides a more rigorous criterion for thermalization than the convergence of the expectation value. This is because the former implies the latter, whereas the reverse is not necessarily true. A similar approach has been employed in previous studies to investigate thermalization dynamics [37, 79–81]. Comprehensive details of how to extract the thermal state corresponding to a global quantum quench are discussed in Section 3.2. In cases where the system undergoes thermalization, DT($t$) is expected to converge to zero in the long time limit.

## 3 Numerical details

### 3.1 Real and imaginary time evolution

The numerical simulations in this study are classified into two distinct categories:

- real time evolution of pure state following a global quench.

- simulation of finite temperature density operators.

For both of these simulation tasks, we employ the MPS-based Time Dependent Variational Principle (TDVP) algorithm [23, 24] with second order integration scheme. This choice affords us a significant advantage over exact diagonalization methods, allowing us to simulate systems of much larger sizes than can be accommodated by the current exact diagonalization techniques.

The quench protocol implemented in this study, with energy rescaled to $|J| = 1$, is as follows: At time $t = 0$, the system is prepared in the ground state of the Hamiltonian $\hat{H}_i(\alpha, 0)$, which is a $\mathbb{Z}_2$ symmetric Greenberger-Horne-Zeilinger (GHZ) state oriented along the longitudinal direction:

$$|\psi_i\rangle = \frac{1}{\sqrt{2}}(|\rightarrow, \ldots \rightarrow, \rightarrow, \rightarrow \ldots, \rightarrow\rangle_x + |\leftarrow, \ldots \leftarrow, \leftarrow, \leftarrow \ldots, \leftarrow\rangle_x), \tag{10}$$

GHZ state is characterized by the PDF $P_l^{\text{GHZ}}(m) = \frac{\delta_{m,|l|/2}}{2}$ which is sharply bimodal with two peaks at $m = l/2$ and $m = -l/2$ respectively. Equation (10) can be explicitly represented as an exact Matrix Product State (MPS) with bond dimension $\chi = 2$. Subsequently, a global

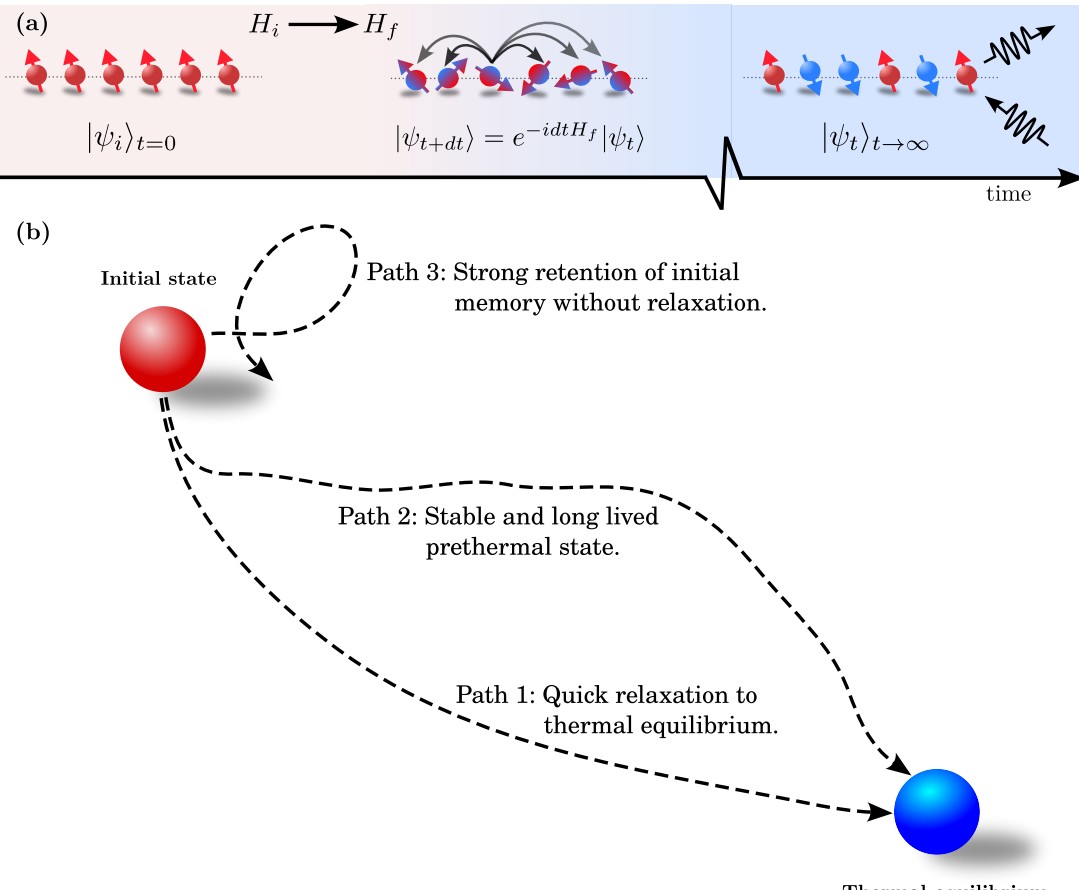

Figure 1: **(a)** Global Quench protocol: System is initialized as the ground state of a trivial hamiltonian $\hat{H}_i$ (in our case the initial state is a Greenberger-Horne-Zeilinger (GHZ) state), at time $t = 0$ the system is suddenly quenched to a final Hamiltonian $\hat{H}_f$ and the initial state is unitarily evolved with the final Hamiltonian. **(b)** The nonequilibrium state following a global quantum quench can exhibit different relaxation behavior; Path 1: a direct relaxation to thermal equilibrium with a single time scale, Path 2: a quick relaxation to a long lived prethermal state eventually followed a relaxation to thermal equilibrium, Path 3: a strong retention of initial memory and suppression of relaxation to thermal equilibrium.

quench is initiated along the transverse field $h$ to a final Hamiltonian $\hat{H}_f(\alpha, h_f)$ and the system is evolved unitarily using the expression $|\psi_{t+dt}\rangle = e^{-idt\hat{H}_f}|\psi_t\rangle$. The evolution is monitored by calculating the Full Counting Statistics (FCS) of the order parameter at each time step. The details of the quench protocol is pictorially represented in Figure 1 **(a)**. The finite temperature density operator is simulated by an imaginary time evolution starting from a maximally mixed state at infinite temperature. Additional details pertaining to the calculation of the thermal density operator are provided in Appendix B.1. For both sets of simulations, we maintain a fixed maximum bond dimension of the MPS at $\chi_{max} = 128$. Furthermore, Trotter time steps of $dt = 0.05$ and $d\beta = 0.001$ are used for real and imaginary time evolution, respectively. There is a finite time-step error of $O(dt^3)$ per time step and $O(dt^2)$ per unit time [82]. In Appendix B.3 we access the accuracy of the TDVP data by comparing the TDVP results with the exact results obtained by the full diagonalization of a system of size $N = 14$. Furthermore, we test the convergence of the data by calculating the relative error for three increasing bond dimensions.

## 3.2 Extraction of effective temperature of a global quench

A global quantum quench $\hat{H}_i(\alpha, 0) \to \hat{H}_f(\alpha, h)$ in an isolated system adds an extensive amount of energy to the system. Consequently, the system relaxes to a state at a higher energy level than the ground state of the post-quench Hamiltonian [70],

$$\lim_{N \to \infty} \frac{1}{N} \frac{\langle \psi_t | \hat{H}_f | \psi_t \rangle}{\langle \psi_t | \psi_t \rangle} > \lim_{N \to \infty} \frac{1}{N} \frac{\langle \psi_0 | \hat{H}_f | \psi_0 \rangle}{\langle \psi_0 | \psi_0 \rangle}, \tag{11}$$

where $|\psi_0\rangle$ is the ground state of the post-quench Hamiltonian $\hat{H}_f(\alpha, h)$. The left hand side of Equation (11) is a conserved quantity because the real time evolution of $|\psi_t\rangle$ is unitary. For every global quantum quench we can attribute an effective temperature $\beta_{\text{eff}}$ which is the temperature at which the thermal energy density above the ground state of the post-quench Hamiltonian matches the conserved energy density of the system,

$$\frac{1}{N} \frac{\langle \psi_t | \hat{H}_f | \psi_t \rangle}{\langle \psi_t | \psi_t \rangle} = \frac{1}{N} \frac{\text{Tr}(\hat{\rho}_\beta \hat{H}_f)}{\text{Tr}(\hat{\rho}_\beta)}. \tag{12}$$

The effective temperature is extracted by solving Equation (12). The left hand side of the equation is trivially calculated as $\langle \psi_t | \hat{H}_f | \psi_t \rangle = \langle \psi_i | e^{it\hat{H}_f} \hat{H}_f e^{-it\hat{H}_f} | \psi_i \rangle = \langle \psi_i | \hat{H}_f | \psi_i \rangle$, and the right hand side can be calculated for a series of $\beta$ by numerically solving Equation (23) and calculating the energy density at each instance. The precision of $\beta_{\text{eff}}$ depends on the trotter steps $d\beta$ in the solution to equation (23). In Fig. 2 we plot the numerical solution of equation (12). The energy density attributed to quench (represented by the black dashed line) in our setup is independent of the post-quench parameters because the spin-spin interaction term in the Hamiltonian (3) is normalized with the Kac normalization (4), whereas the expectation value $h \langle \psi_i | \sum_j \hat{s}_j^z | \psi_i \rangle$ taken over the transverse field term is trivially zero. If we extend the simulation to a larger $\beta$ (i.e., lower temperature), all curves will converge to the ground state energy density of $\hat{H}_f$ at the corresponding post-quench parameters. Once $\beta_{\text{eff}}$ is extracted, we can calculate the corresponding thermal PDF, $P^{\text{TH}}(m) = P^{\beta_{\text{eff}}}(m)$, using equation (8).

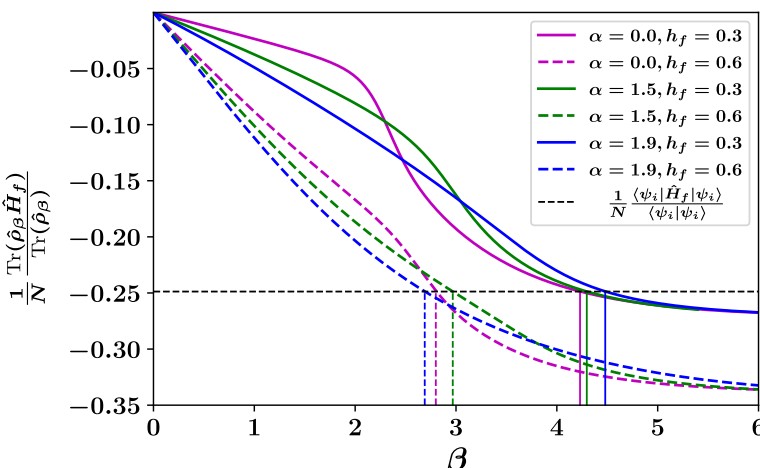

Figure 2: Numerical extraction of $\beta_{\text{eff}}$ corresponding to a global quantum quench. The horizontal black dashed line represent the energy density attributed to the quench. The colored lines represents the energy density as the function of inverse temperature $\beta$ for the corresponding post-quench parameter (in legend). The point at which the colored lines intersects the black dashed lines represents $\beta_{\text{eff}}$ for the corresponding post-quench parameters (represented by vertical colored lines).

# 4 Results

The global quantum quench, as discussed in Section 3.1, induces a dynamical quantum phase transition (DQPT) [83, 84] in LRIM, which has garnered extensive attention in recent years. This transition falls into two distinct categories: the first, known as DQPT-I, is characterized by distinctive behaviors in the time-averaged local order parameter following a global quench across the dynamical critical point [2, 54, 64], and the second, DQPT-II, is marked by non-analytic cusps in the Loschmidt echo rate [3, 67–69, 85]. In LRIM, the dynamical critical points for DQPT-I and DQPT-II coincide at approximately $h_c^{\text{dyn}} \approx 0.5$ for $\alpha \leq 2$ [64, 85]. When $h_f < h_c^{\text{dyn}}$, the system is in the dynamical ferromagnetic phase, which strongly retains the ferromagnetic order of the initial GHZ state following a global quantum quench. This is evident from the persistent oscillation of $P_l(m, t)$ around $P_l^{\text{GHZ}}(m)$. Conversely, when $h_f > h_c^{\text{dyn}}$, the system transits to the dynamical paramagnetic phase, characterized by the rapid dissolution of the initial ferromagnetic order of the initial GHZ state following a global quantum quench. This is signified by the Gaussification of $P_l(m, t)$ [54, 64]. The comprehensive dynamical phase diagram and universality behavior related to the dynamical phase transition in LRIM remain active areas of investigation [66, 86–88].

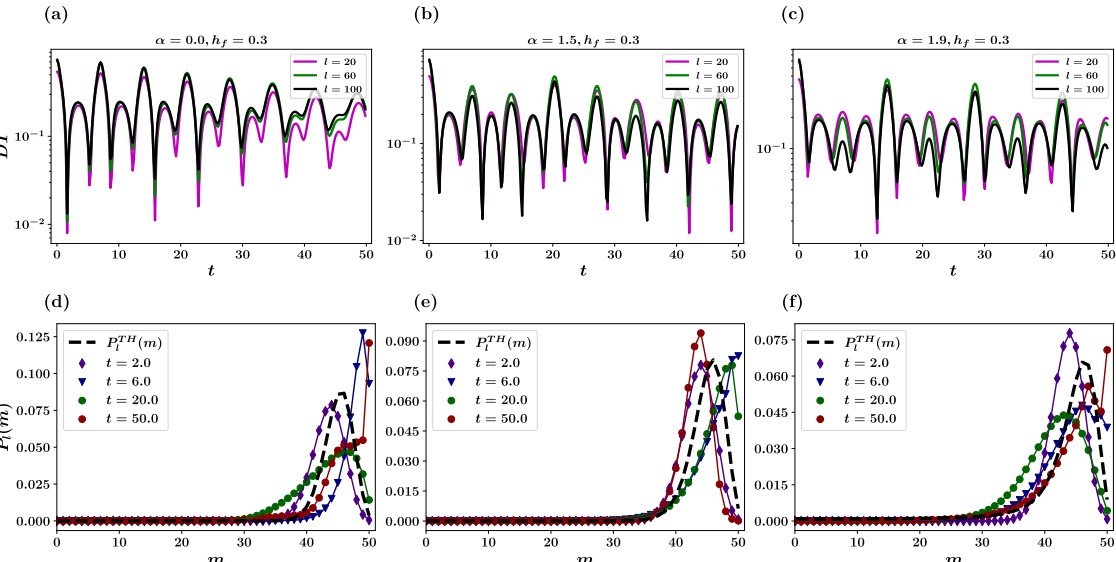

Figure 3: First row: Time evolution of the metric DT($t$) following a global quantum quench to three interaction strength values $\alpha \in \{0.0, 1.5, 1.9\}$ and transverse field $h_f = 0.3$ at three different subsystem sizes $l = \{20, 60, 100\}$. All three points are in dynamical ferromagnetic phases [54, 64]. Second row: $P_l(m)$ versus $m$ for $m \in [0, l/2]$ with $l = 100$ at four different time slices $t = \{2, 6, 20, 50\}$. $P_l(m)$ versus $m$ for $m \in [-l/2, 0)$ is its mirror image. The black dashed curve represents the thermal PDF, $P^{\text{TH}}(m)$ attributed to the corresponding global quantum quenches.

Our primary objective is to examine the convergence behavior of the metric DT($t$) in two distinct dynamical phases of the LRIM. The convergence of DT($t$) towards zero is an indicator of thermalization within a particular phase under consideration. We initialize the system as $\mathbb{Z}_2$ symmetric GHZ state presented in Equation (10), which represents the ground state of the Hamiltonian given in Equation (3) at $h = 0$. This choice is made because the model (3) undergoes a thermal transition from a paramagnetic phase, characterized by a Gaussian probability density function (PDF) at high temperatures, to a ferromagnetic phase with a $\mathbb{Z}_2$ symmetric bimodal PDF, as comprehensively detailed in Appendix C. We maintain that $\alpha < 2$ is

crucial as this region exhibits an interesting landscape encompassing both finite temperature phase transitions [89, 90] and dynamical phase transitions [2, 3, 54, 64–69]. Specifically, we consider three distinct values of interaction strength, namely $\alpha = 0.0, 1.5, 1.9$. At $\alpha = 0.0$, the system exhibits integrability because of its full connectivity and complete permutation symmetry, thereby leading to a lack of thermalization [17]. On the other hand, the choices of $\alpha = 1.5$ and $\alpha = 1.9$ are motivated by the relatively faster equilibration and Gaussification of the PDF following a quench in the dynamical paramagnetic phase, as previously observed [54].

## 4.1 Quench to dynamical ferromagnetic regime

Figure 3 shows the temporal evolution of DT($t$) following a global quantum quench of the transverse field to $h_f = 0.3$ with $\alpha = 0.0, 1.5, 1.9$ for subsystem sizes $l = 20, 60, 100$. Notably, all these points belong to the dynamical ferromagnetic phase [54, 64]. For all three quenches, a persistent oscillation in DT($t$) is evident, indicating that the initial ferromagnetic order is strongly retained and thermalization is suppressed. This behavior aligns with the relaxation mode represented by Path 3 in figure 1(b). Specifically, $\alpha = 0.0$ is in the integrable regime; therefore, thermalization is expected to be absent [26] whereas we anticipate thermalization for quenches with $\alpha = 1.5$ and $\alpha = 1.9$. The apparent suppression of thermalization can be attributed to the confinement behavior. The long-range interaction of the model effectively confines low-energy domain wall kinks into heavier quasiparticles that typically travel slower than free quasiparticles, thereby suppressing the spread of correlations in the system [44, 45]. Consequently, thermalization is still expected but only at significantly longer time scales [47]. Appendix D details the confinement behavior in LRIM where we observe the spreading of connected correlation function $\left\langle \hat{s}_k^x \hat{s}_{k+\Delta}^x \right\rangle_c$ for $\alpha = 1.9$ and $h_f = 0.3$ shows a strong temporal suppression. In figure 3(d), (e), and (f), the colored scattered plots depict $P_l(m)$ as a function

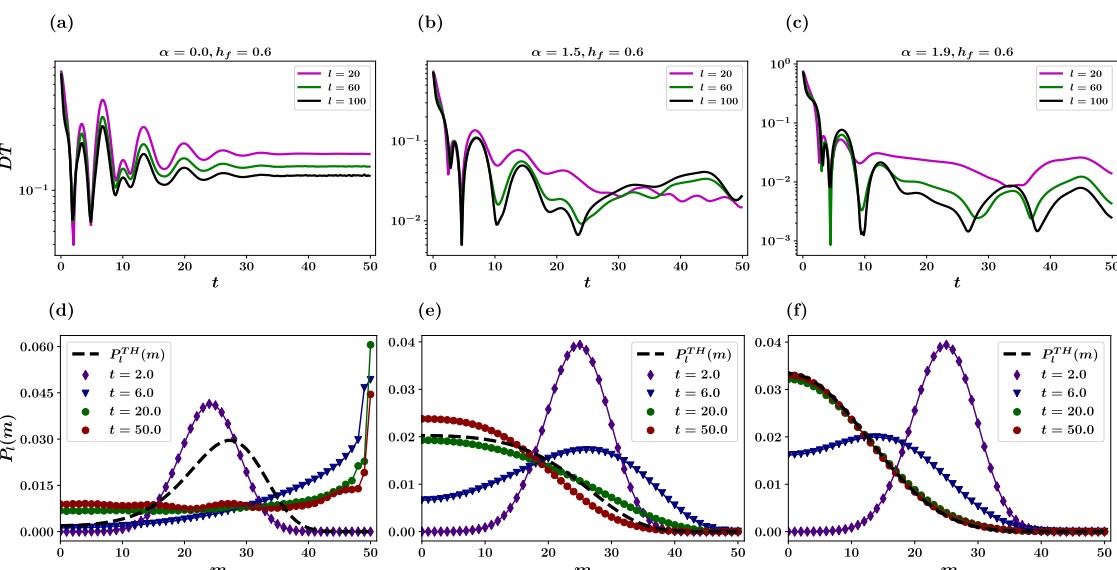

Figure 4: First row: Time evolution of the metric DT($t$) following a global quantum quench to three interaction strength values $\alpha \in \{0.0, 1.5, 1.9\}$ and transverse field $h_f = 0.6$ at three different subsystem sizes $l = \{20, 60, 100\}$. All three points are in dynamical paramagnetic phases [54, 64]. Second row: $P_l(m)$ versus $m$ for $m \in [0, l/2]$ with $l = 100$ at four different time slices $t = \{2, 6, 20, 50\}$. $P_l(m)$ versus $m$ for $m \in [-l/2, 0)$ is its mirror image. The black dashed curve represents the thermal PDF, $P^{\text{TH}}(m)$ attributed to the corresponding global quantum quenches.

of $m$ at four distinct time slices. The black dashed curve represents the Probability Density Function (PDF) of the expected thermal state, $P_l^{\text{TH}}(m)$. We observe that the time-evolving $P_l(m)$ oscillates persistently around $P_l^{\text{TH}}(m)$. Of particular importance is the observation that, in all three cases, the thermal PDFs are bimodal, indicating the presence of long-range ferromagnetic order. This observation suggests that if the system eventually thermalizes for these post-quench parameters at extended time scales, it would exhibit a long-range ferromagnetic order. This finding further strengthens the argument that this is indeed a dynamical ferromagnetic phase.

## 4.2 Quench to dynamical paramagnetic regime

Figure 4 illustrates the temporal evolution of DT$(t)$ following a global quantum quench of the transverse field to $h_f = 0.6$, with $\alpha = 0.0, 1.5, 1.9$ for subsystem sizes $l = 20, 60, 100$. These points are located within the dynamical paramagnetic phase [54, 64]. Notably, these quenches exhibit a distinct relaxation behavior of DT$(t)$ compared with the previous cases. In Figure 4(a), we observe rapid equilibration for all values of $l$. However, it is essential to highlight that DT$(t)$ remains at or above the order of $O(10^{-1})$ following equilibration, which suggests a lack of thermalization. This behavior aligns with expectations, because $\alpha = 0$ represents an integrable point. For $\alpha = 1.5$, DT$(t)$ does not exhibit stable equilibration (see figure 4 (b)); Finally, when $\alpha = 1.9$, we observe equilibration for $l = 60, 100$ (see figure 4 (c)). DT$(t)$ exhibits a stable oscillation around a constant value of approximately $O(10^{-3})$. A more comprehensive picture is shown in Fig. 4(f), where the late-time PDF perfectly overlaps with the corresponding thermal PDF represented by a black dashed curve. This is indicative of thermalization of the corresponding quench. Although we observed signatures of thermalization, the system is not in a de-confined phase [4,91]. In Appendix D, the connected correlation function $\left\langle \hat{s}_k^x \hat{s}_{k+\Delta}^x \right\rangle_c$ for $\alpha = 1.9$ and $h_f = 0.6$ still exhibits weaker temporal suppression. A recent study observed a de-confinement transition for a system of up to 31 spins for a much higher value of the transverse field [4]. This suggests that while strong confinement suppresses thermalization, signatures of thermalization can still be detected in the presence of weak confinement.

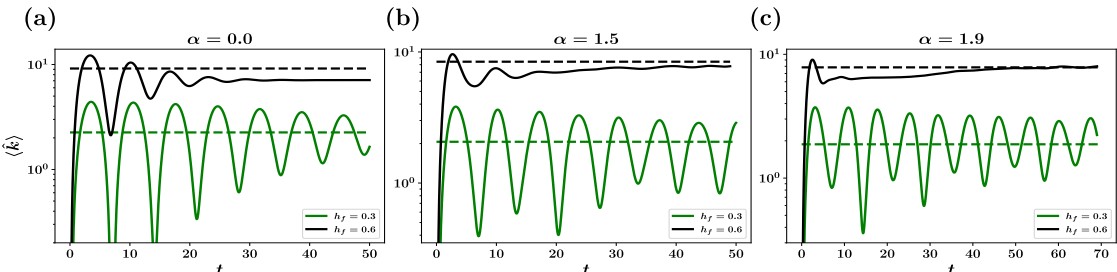

Figure 5: Time evolution of domain wall kinks, $\langle \hat{k} \rangle$, following a global quantum quench to three interaction strength values $\alpha \in \{0.0, 1.5, 1.9\}$ (Panels (a), (b), and (c) respectively) and $h_f = 0.3$ and $h_f = 0.6$. The dashed horizontal lines represent the expected thermal value of domain wall kinks, $\langle \hat{k} \rangle^{\text{TH}}$ corresponding to the quenches.

To further support this observation we study the post quench temporal evolution of domain wall kinks defined as,

$$\hat{k} = \sum_{j=1}^{l-1} \frac{1 - \hat{s}_i^x \hat{s}_{i+1}^x}{2}. \tag{13}$$

$\hat{k}$ counts the number of nearest neighbor kinks in the $\hat{x}$ direction within subsystem $l$. Because

confinement bounds the domain walls kinks into heavier quasiparticles, it is a relevant parameter to study. In Figure 5, we illustrate the temporal evolution of the average domain-wall kinks, $\langle \hat{k} \rangle$, following a global quantum quench. As anticipated, quenches to the dynamical ferromagnetic phase with $h_f = 0.3$ display persistent oscillations around the thermal value, indicating a lack of thermalization. Conversely, for quenches to the dynamical paramagnetic phase with $h_f = 0.6$, we observe distinctly different post-quench behavior. In the case of $\alpha = 0$, the domain wall kinks equilibrate to a stable value that differs from the expected thermal value, as expected because it is an integrable point. This observation complements the post-quench behavior of DT, as depicted in Figure 4 (a). Although $\alpha = 1.5$ is a non-integrable point, thermalization is not observed within the simulation time. At later times, a stable prethermal plateau, close but distinct from the expected thermal value, becomes apparent. Conversely, for a quench corresponding to $\alpha = 1.9$, the average domain wall kinks converge to the expected thermal value. Notably, before reaching the thermal value, the kink density exhibits a relatively stable prethermal plateau until time $t \simeq 35$. This relaxation mode, which is characterized by two time scales, is represented by Path 2 in Figure 1(b). This discovery provides another robust indicator of thermalization in weakly confined regimes.

## 5 Conclusion

We investigate the relaxation dynamics of the long-range Ising model subsequent to a global quantum quench of the transverse field, assessing the thermalization on a computationally viable time scale according to the canonical Gibbs ensemble (CGE). The model is non-integrable for all values of $\alpha$ except at the extremes ($\alpha = 0.0, \infty$), where we anticipate thermalization following a global quantum quench. However, the long-range Ising model exhibits confinement, which suppresses correlation spreading and eventually suppresses thermalization. Starting from the Greenberger-Horne-Zeilinger (GHZ) state, we quench the system to two distinct dynamical regimes. As anticipated, robust confinement suppresses thermalization for smaller quenches, specifically in the dynamical ferromagnetic region, where the metric DT exhibits persistent oscillations characteristic of the masses of bound mesons. Conversely, for quenches to the dynamical paramagnetic region, a notably different behavior emerges. The persistent oscillation diminishes and the DT relaxes more rapidly. Although conclusive evidence of thermalization for $\alpha = 1.5$ is not observed within the simulation time, compelling indications of the thermalization surface for $\alpha = 1.9$ are based on the relaxation of DT. This observation gains additional support from the convergence of the domain wall kinks to the expected thermal value.

## Acknowledgements

Nishan Ranabhat thanks Alvise Bastianello for fruitful discussion and suggesting the future extension of the work. The numerical simulation of this project was performed at the Ulysses v2 cluster at SISSA.

## A Exact results for smaller systems

For small systems we can calculate the time evolution of relevant order parameters by exact diagonalization of the post-quench Hamiltonian. We begin from our initial state, $\mathbb{Z}_2$ symmetric GHZ state,

$$|\psi_0\rangle = \frac{1}{\sqrt{2}}(|\rightarrow, \ldots \rightarrow, \rightarrow, \rightarrow \ldots, \rightarrow\rangle + |\leftarrow, \ldots \leftarrow, \leftarrow, \leftarrow \ldots, \leftarrow\rangle). \tag{14}$$

The time evolved state is given by $|\psi_t\rangle = e^{-i\hat{H}t}|\psi_0\rangle$, where $\hat{H}$ is the post-quench Hamiltonian 3. We proceed by expanding $|\psi_0\rangle$ in the eigenbasis, of the post-quench Hamiltonian,

$$|\psi_0\rangle = \sum_{j=0}^{2^N-1} q_j |E_j\rangle, \tag{15}$$

where $q_j = \langle E_j | \psi_0 \rangle$. Further expanding $|E_j\rangle$ in the computational basis, $|E_j\rangle = \sum_n c_n^j |n\rangle$, we can derive the expression for $q_j$ as,

$$q_j = \langle E_j | \psi_0 \rangle = \frac{\left(c_{|\rightarrow,\ldots,\rightarrow\rangle}^j\right)^* + \left(c_{|\leftarrow,\ldots,\leftarrow\rangle}^j\right)^*}{\sqrt{2}}. \tag{16}$$

The post-quench state is

$$|\psi_t\rangle = \sum_n X_n(t) |n\rangle, \tag{17}$$

where $X_n(t) = \sum_{j=0}^N q_j c_n^j e^{-iE_j t}$. We can now calculate the time evolution of the expectation value of a generic parameter $\hat{O}$ as,

$$O(t) = \langle \psi_t | \hat{O} | \psi_t \rangle = \sum_{n,\tilde{n}} X_{\tilde{n}}^{\dagger}(t) X_n(t) \langle \tilde{n} | \hat{O} | n \rangle. \tag{18}$$

If $|n\rangle$ is the simultaneous eigenket of the order parameter $\hat{O}$ then 18 becomes,

$$O(t) = \sum_n |X_n(t)|^2 O_n. \tag{19}$$

Finally, with the full eigenvalues of hamiltonian at hand we can also calculate the energy density corresponding to a thermal density matrix $\hat{\rho}_\beta$,

$$\epsilon_\beta = \frac{\sum_j E_j e^{\beta E_j}}{\sum_j e^{\beta E_j}}. \tag{20}$$

## B  Simulations details

In this section we present the details of numerical simulation complementary to the results in the main text.

### B.1  Simulation of finite temperature density operator

The finite temperature states can be simulated by casting the density operator as locally purified tensors [92, 93]. The thermal density operator is defined by Gibbs distribution $\hat{\rho}_\beta = \frac{e^{-\beta\hat{H}}}{Tr[e^{-\beta\hat{H}}]}$ where $\beta = \frac{1}{T}$ is the inverse temperature. At $\beta = 0$ (infinite temperature) the state is maximally mixed and is given as the tensor product of local identities $\hat{\rho}_0 = \bigotimes_{i=1}^N \mathbf{1}^{\sigma_i',\sigma_i} = \mathbb{1}$, where each $\mathbf{1}^{\sigma_i',\sigma_i}$ is a unit matrix of size $(d,d)$, i.e. $\mathbf{1}^{\sigma_i',\sigma_i} = [\delta_{\sigma_i',\sigma_i}]_{d\times d}$

and $d$ is the dimension of the local Hilbert space (for spin $\frac{1}{2}$, $d = 2$). The density operator for any finite temperature (non-zero $\beta$) is

$$\hat{\rho}_\beta \propto e^{-\beta \hat{H}} = e^{-\frac{\beta}{2}\hat{H}} \mathbb{1} e^{-\frac{\beta}{2}\hat{H}} \tag{21a}$$

$$\propto e^{-\frac{\beta}{2}\hat{H}} \hat{\rho}_0 e^{-\frac{\beta}{2}\hat{H}}. \tag{21b}$$

We keep the density operator operator in locally purified form $\hat{\rho} = \mathbb{X}\mathbb{X}^\dagger$ at each stage where $\mathbb{X}$ is represented as tensor

$$\mathbb{X}^{\sigma_1,\sigma_2,...\sigma_i,...,\sigma_N}_{k_1,k_2,...,k_i,...,k_N} = \mathbf{X}^{\sigma_1,k_1}_{c_0,c_1}\mathbf{X}^{\sigma_2,k_2}_{c_1,c_2}\ldots\mathbf{X}^{\sigma_i,k_i}_{c_{i-1},c_i}\ldots\mathbf{X}^{\sigma_N,k_N}_{c_{N-1},c_N}, \tag{22}$$

where $\sigma_i = d$, $k_i = d$ are the physical index and the Kraus index, they remain fixed, and $1 \le c_i \le \chi_{max}$ is the bond index and $\chi_{max}$ is the maximum value of bond dimension. The density operator initialized at infinite temperature can now be purified to a finite temperature in trotterized steps

$$\hat{\rho}_{\beta+d\beta} = e^{-\frac{d\beta}{2}\hat{H}}\hat{\rho}_\beta e^{-\frac{d\beta}{2}\hat{H}} \tag{23a}$$

$$= e^{-\frac{d\beta}{2}\hat{H}}\mathbb{X}\mathbb{X}^\dagger e^{-\frac{d\beta}{2}\hat{H}} \tag{23b}$$

$$= e^{-\frac{d\beta}{2}\hat{H}}\mathbb{X}[e^{-\frac{d\beta}{2}\hat{H}}\mathbb{X}]^\dagger. \tag{23c}$$

Equation (23) can be simulated using imaginary time TDVP ($-idt \to -d\beta$) in only the half section of the density operator operator and never contracting the $X$ and $X^\dagger$ layer during the evolution, thus strictly preserving the locally purified form.

$$\hat{\rho}_{\beta=0} \propto \bigotimes_{i=1}^{N} \mathbb{1}^{\sigma_i',\sigma_i} = \quad\begin{array}{ccc} \sigma_1 & \sigma_i & \sigma_N \\ | & | & | \\ | \quad \otimes \ldots \otimes & | \quad \otimes \ldots \otimes & | \\ | & | & | \\ \sigma_1' & \sigma_i' & \sigma_N' \end{array}$$

Figure 6: Maximally mixed density operator at $\beta = 0$ as the tensor product of local identities.

Figure 6 shows the tensor notation of the infinite temperature density operator $\hat{\rho}_{\beta=0}$ which is a tensor product of identity matrices of size $(d, d)$, where $d$ is the physical dimension. Rather than working with the density operator as an MPO we represent the density operator in the locally purified form [93, 94] which is positive semi-definite by construction and keep it in locally purified form at every stage of the thermal purification process. In Fig. 7 we represent $\hat{\rho}_{\beta=0}$ in the locally purified form $\mathbb{X}_{\beta=0}\mathbb{X}^\dagger_{\beta=0}$, where the index in purple is an auxiliary index called the Krauss index.

we can now evolve one of the halves ($\mathbb{X}$ or $\mathbb{X}^\dagger$) as shown in equation (23) and the evolution on the other half is its trivial conjugate. This approach is computationally efficient as we can work with cheaper MPS instead of more expensive MPDO. In Fig. 8 one half of the $\hat{\rho}_{\beta=0}$ in locally purified form is shown, form here on we will only work with this half.

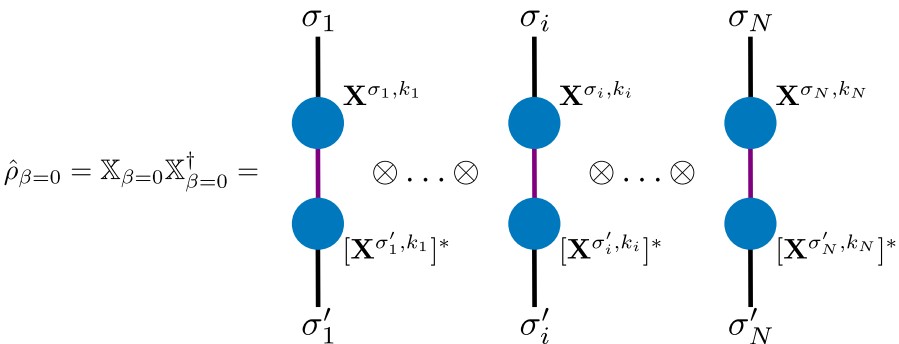

Figure 7: Representing $\hat{\rho}_{\beta=0}$ in the locally purified form.

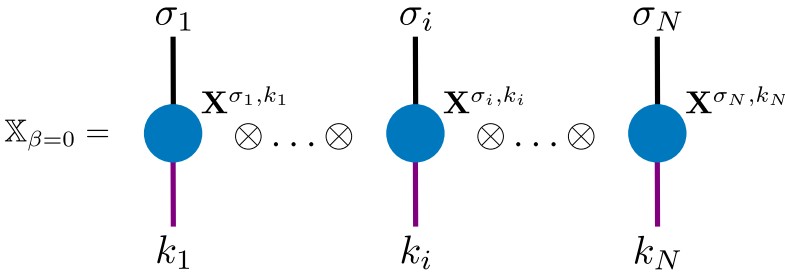

Figure 8: One half of the $\hat{\rho}_{\beta=0}$ in the locally purified form.

Algebraically, $\mathbb{X}_{\beta=0}$ can be written as

$$\mathbb{X}^{\sigma_1,k_1,\ldots\sigma_i,k_i,\ldots,\sigma_N,k_N} = \mathbf{X}^{\sigma_1,k_1} \otimes \ldots \mathbf{X}^{\sigma_i,k_i} \ldots \otimes \mathbf{X}^{\sigma_N,k_N} \,. \tag{24}$$

For the system of spin one-half particles we choose $\mathbf{X}$ as

$$\mathbf{X}^{\sigma_i,k_i} = \frac{1}{\sqrt{2}}\begin{pmatrix} 1 & 0 \\ 0 & 1 \end{pmatrix}, \qquad \forall\, i \in \{1,2,\ldots,N\}\,, \tag{25}$$

as shown in Fig. 9. This particular choice is taken to preserve the trace of the density operator,

$$\mathbf{X}^{\sigma_i,k_i} = \raisebox{-1.5em}{\includegraphics{}} = \frac{1}{\sqrt{2}}\begin{pmatrix} 1 & 0 \\ 0 & 1 \end{pmatrix}$$

Figure 9: Choice of $\mathbf{X}^{\sigma_i,k_i}$ to preserve the trace of $\hat{\rho}$.

$$\sum_k \mathbf{X}^{\sigma,k}[\mathbf{X}^{\sigma',k}]^* = \frac{1}{2}\begin{pmatrix} 1 & 0 \\ 0 & 1 \end{pmatrix}. \tag{26}$$

Finally, we reshape $\mathbb{X}_{\beta=0}$ from a string of $2 \times 2$ matrices to a string of four legged tensors of shape $(1,2,2,1)$ as shown in Fig. 10, which is a MPS of bond dimension 1.

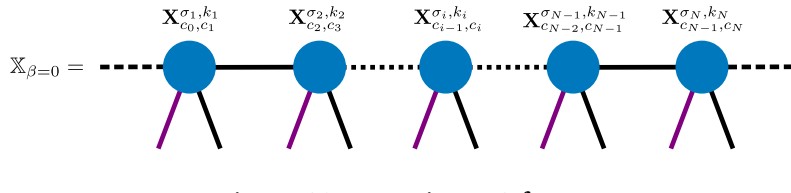

Figure 10: $\mathbb{X}_{\beta=0}$ in MPS form.

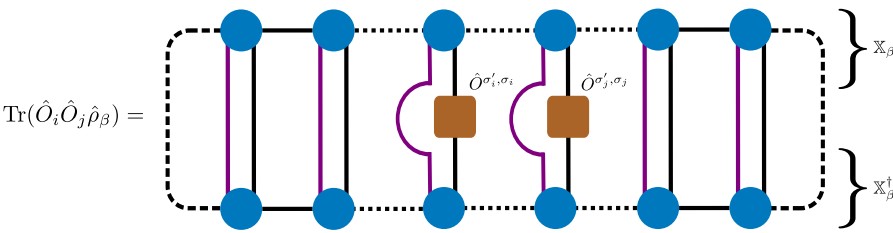

Figure 11: Expectation of the local operator $\hat{O}_i$ in thermal density operator $\hat{\rho}_\beta$.

Now that we have our initial state as an MPS, we can simulate a finite temperature density operator by solving the equation (23),

Numerically, equation (23) can be solved for long-range spin systems through imaginary time evolution, (where $idt$ is transformed into $d\beta$) using the Time-Dependent Variational Principle (TDVP). The TDVP algorithm employed for simulating the thermal state remains fundamentally identical to that used for the real-time evolution of the pure state, with the distinction of an additional auxiliary Krauss index. However, in the thermal purification of a closed system, the Krauss index becomes obsolete because all physical operators act solely on the physical index, and the Krauss indices are contracted among themselves [93]. Figure 11 illustrates the tensor network diagram for computing the expectation value of a two point operator $\hat{O}_i\hat{O}_j$ acting on site $i$ and $j$ within the thermal state $\hat{\rho}_\beta$.

## B.2 Calculating full counting statistics with MPS

The Central object in the calculating the full probability distribution function of an order parameter is the moment generating function $G_l(\theta, t) = \text{Tr}(\hat{\rho}_t e^{i\theta \hat{M}_l})$. For pure state, the density matrix can be written as $\hat{\rho}_t = |\phi_t\rangle\langle\phi_t|$ such that $G_l(\theta, t) = \langle\phi_t|e^{i\theta\hat{M}_l}|\phi_t\rangle = \langle\phi_t|\prod_{j=i}^{i+l-1} e^{i\theta\hat{s}_j^x}|\phi_t\rangle$. The state $|\phi_t\rangle$ can be represented as a matrix product state (MPS) [95], and the single-site operator $e^{i\theta\hat{s}_j^x}$ can be expressed as a two-by-two matrix. Utilizing this representation, the moment generating function can be computed by sandwiching the operators between the matrix product states, as depicted in Figure 12. By obtaining $G_l(\theta, t)$, the complete probability distribution $P_l(m, t)$ is computed numerically by discretizing the Fourier integral in equation 8.

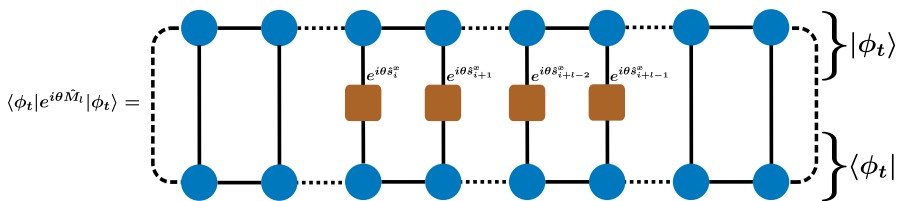

Figure 12: Computing the generating function $G_l(\theta, t)$ in matrix product state representation. The site $i$ is chosen such that the subsystem of size $l$ is in the center of the full system.

## B.3 Errors and data convergence

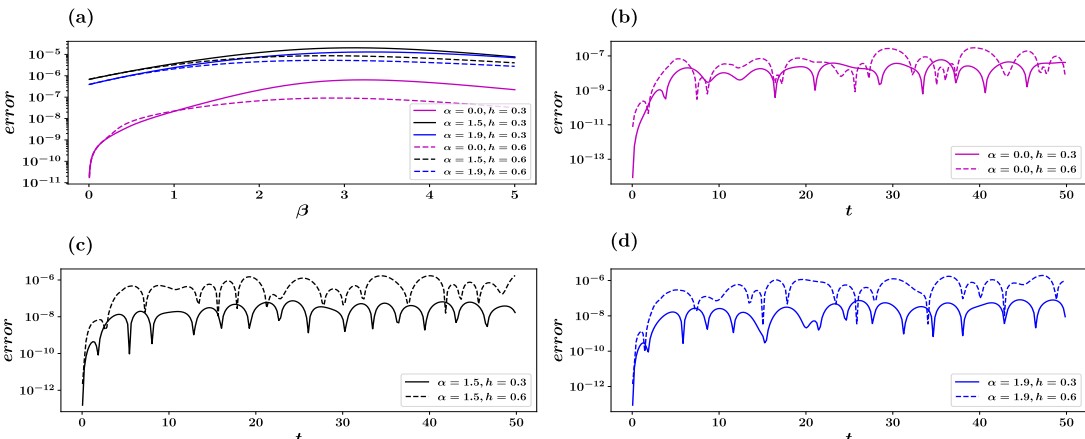

Figure 13: Absolute error in the energy density, $|\epsilon_\beta^{\mathrm{ED}} - \epsilon_\beta^{\mathrm{TDVP}}|$, of thermal states - **(a)**. Absolute errors in the evolution domain wall kinks $|\langle\hat{k}\rangle^{\mathrm{ED}} - \langle\hat{k}\rangle^{\mathrm{TDVP}}|$ following a quantum quench - **(b),(c),(d)**. The numerically exact results are calculated using equations 19 and 20 as detailed in A. TDVP results are obtained with bond dimension $\chi = 128$. The system size considered is N=14.

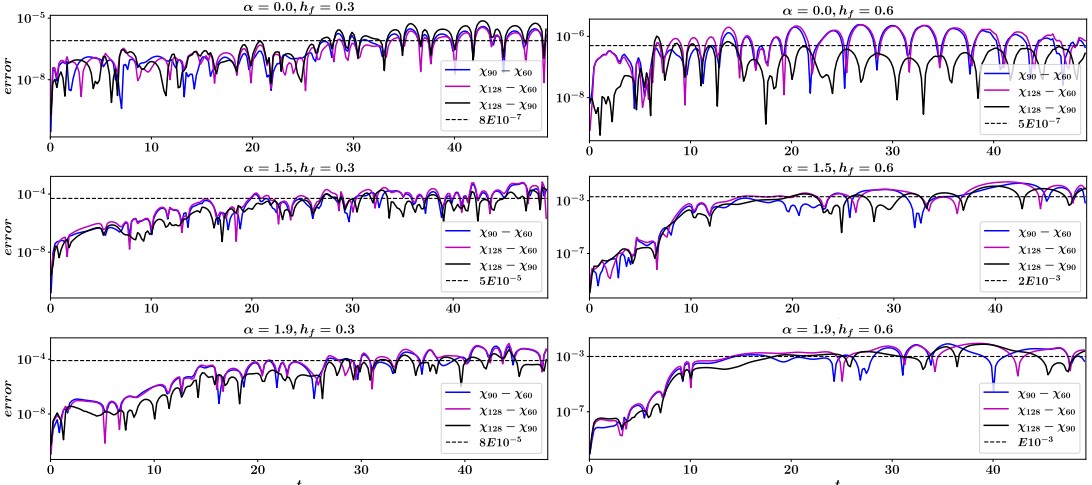

Figure 14: Convergence of the TDVP data for $\mathrm{DT}(t)$ with increasing bond dimensions, $\chi = 60, 90, 128$, for six different post-quench parameters considered in the main text. The black dashed line is for visual guidance.

We conducted two types of error analysis to assess the accuracy of the numerical results. In figure 13, we assess the absolute error of the TDVP algorithm in comparison with the numerically exact full diagonalization results for a system with size $N = 14$ and various post-quench parameters. Figure 13, panel **(a)**, shows the absolute error in the energy density of the thermal states, defined as $|\epsilon_\beta^{\mathrm{ED}} - \epsilon_\beta^{\mathrm{TDVP}}|$. The absolute error remains of the order $O(10^{-5})$ or smaller across the entire temperature range under consideration. Figures 13, **(b)**, **(c)**, and **(d)** show the absolute error in domain wall kinks, defined as $|\langle\hat{k}\rangle^{\mathrm{ED}} - \langle\hat{k}\rangle^{\mathrm{TDVP}}|$, following a quantum quench to various post-quench parameters. $\langle\hat{k}\rangle$ is defined in equation 13 and the computational basis $\{|n\rangle\}$ is its simultaneous eigenbasis. Notably, the error rapidly converge and is of order $O(10^{-6})$ or smaller for all the cases studied.

In Figure 14, we investigate the convergence of the TDVP data for $DT(t)$ by computing the relative errors $|DT^{\chi_1}(t) - DT^{\chi_2}(t)|$ for three increasing bond dimensions. Our observations reveal that the relative error eventually converges and consistently remains in the order $O(10^{-3})$ or smaller for all cases. It is noteworthy that the error for $\alpha = 0.0$ is several orders of magnitude smaller than that for the other values of $\alpha$. This is attributed to $\alpha = 0.0$ being an integrable point with an extensive number of conserved quantities, and therefore has a smaller Hilbert space to be explored compared to non-integrable points. Furthermore, for $\alpha = \{1.5, 1.9\}$, the error for $h_f = 0.3$ is approximately two orders of magnitude smaller than that for $h_f = 0.6$. This discrepancy arises because the former case exhibits dynamical confinement, which effectively suppresses the spread of correlations and constrains the total Hilbert space that can be explored during time evolution.

## C Thermal phase transition in long range Ising model

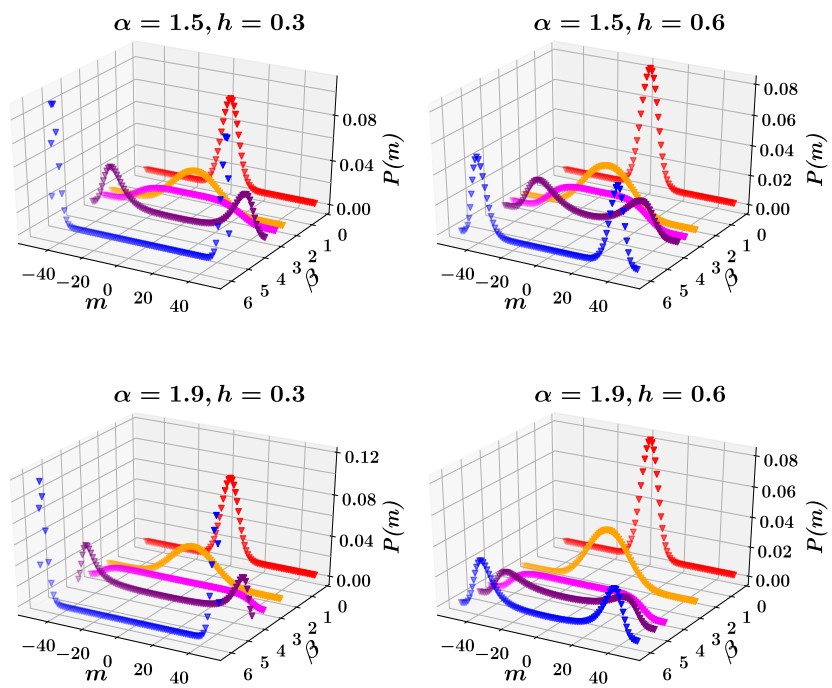

Figure 15: Thermal phase transition of long range Ising model at four different points in parameter space. The initial state in all cases is the maximally mixed state at infinite temperature represented by $\hat{\rho}_{\beta=0}$, refer to 6. The color coding from red to blue signifies decreasing temperature.

For values of $\alpha > 2$, the long-range Ising model falls within the regime of short-range interactions and does not exhibit any finite-temperature phase transitions [89]. Extensive investigations into the critical properties of the thermal phase transition in the quantum long-range Ising model have been conducted using numerically exact path integral Monte Carlo methods [90]. The thermal phase transition is qualitatively depicted in Figures 15 for specific parameter values: $\alpha = 1.5, 1.9$ and $h = 0.3, 0.6$. As described in Section B.1, the simulation begins with a maximally mixed state at $\beta = 0$. This initial state is characterized by a sharply peaked Gaussian distribution of $P(m)$ centered around $m = 0$, which signifies a strongly para-

magnetic phase. As the system is gradually cooled by increasing $\beta$, the distribution gradually widens, eventually becoming nearly flat around the critical temperature. A further reduction in temperature leads to the emergence of a bimodal distribution of $P(m)$, which is indicative of the ferromagnetic phase. Notably, this transition from a unimodal Gaussian distribution to a bimodal distribution highlights the $\mathbb{Z}_2$ symmetry that is inherent in the long-range Ising Hamiltonian.

## D  Confinement dynamics in different regimes

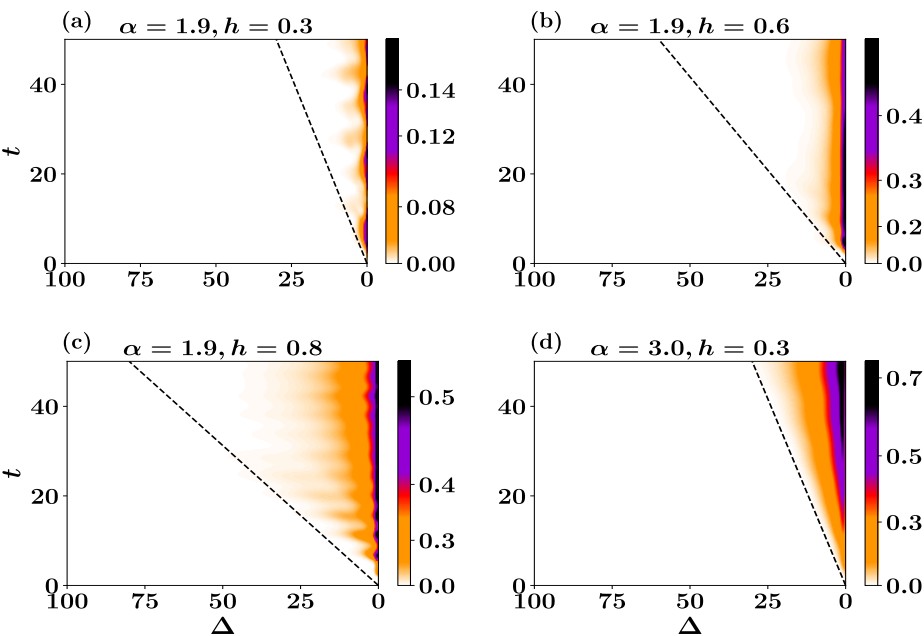

Figure 16: Real time dynamics of half chain connected correlation function $\left\langle \hat{s}_k^x \hat{s}_{k+\Delta}^x \right\rangle_c$ after a global quantum quench of the transverse field starting from a fully polarized initial state. The dashed black lines is $v_{max} = 2h$ line for nearest neighbor transverse field Ising model [42].

Confinement phenomena in the long-range Ising model result from ferromagnetic interactions extending over long distances between the interacting spins. However, the strength of confinement varies within different regions of the phase space [44, 45]. In this section, we present comprehensive numerical results pertaining to the temporal spreading of correlations in the long-range Ising chain following a sudden quench to various post-quench Hamiltonians starting from a fully polarized initial state denoted as $|\psi_i\rangle = |\leftarrow, \leftarrow, \ldots, \leftarrow, \ldots, \leftarrow, \leftarrow\rangle_x$.

Figure 16 illustrates the time evolution of the half chain connected correlation function $\left\langle \hat{s}_k^x \hat{s}_{k+\Delta}^x \right\rangle_c = \left\langle \hat{s}_k^x \hat{s}_{k+\Delta}^x \right\rangle - \left\langle \hat{s}_k^x \right\rangle \left\langle \hat{s}_{k+\Delta}^x \right\rangle$ in a chain of 200 spins, where $k$ is kept fixed at the center of the chain. In panels (a), (b), and (c), we examine a fixed value of $\alpha = 1.9$ while varying the transverse field $h = 0.3, 0.6, 0.8$. Notably, panel (a) shows a pronounced signature of confinement, which gradually diminishes as the value of $h$ increases, as shown in panels (b) and (c). This behavior is expected because the transverse field competes with long-range interactions and weakens the confinement effect. In panel (d), we observe a linear light cone spreading of the correlation with the maximum possible velocity, $v_{\max} = 2h$ [42].

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
