# Peer review of "Thermalization of long range Ising model in different dynamical regimes: a full counting statistics approach"

_SciPost Physics Core, doi:SciPost Phys. Core 7, 017 (2024)_

## Round 2 · Referee Report · Anonymous · 2023-3-10

Report
In this work the authors study thermalization in two different dynamical regimes of the long range transverse field Ising model after a global quench. To determine whether or not the system is thermalizing they follow the time evolution of the distance of the system to the thermal state. Since this approach relies on the full probability distribution function rather than the expectation value of one or several observables, which is often the case in other studies, it should provide better results. However, as the authors themselves note, this is not the first time this approach has been used. Thus this is a case of applying an already established method to this particular model. Furthermore, the results also seem to mostly confirm what has previously been known from other studies. The only possible exception here is the observation that the metric considered in this paper shows thermalization at weak confinement. However, this is left unexplored. Based on this, I do not believe that the paper, in its current form, satisfies SciPosts acceptance criteria and additional work should be put towards understanding the relation between confinement and thermalization before considering the paper for publication.
Regarding the presentation of the paper, it leaves a lot to be desired. Not only does the text contain many grammatical errors, but the main results of the work are not conveyed very clearly. I would recommend the authors more clearly emphasize the novel results presented in this work. Additionally, a large part of the text is assigned to describing TDVP. While this would typically be relegated to the Appendix I see no issue with this, however, it may be useful to do something similar for the model. Namely, the authors often refer to the different dynamical regimes present here. Perhaps a few sentences could be added briefly describing the previous works and what is already known about these regimes. This would also make the point of the current work clearer to non-experts. Finally, it would also be beneficial if Figures 2 and 3 were moved to the main text, where they are referenced.
Moving on, this work relies exclusively on numerical simulations of finite systems using TDVP. While the authors do provide plots comparing different system size to show that their results are not significantly affected by finite size effects (at least at the larger system sizes L=60 and L=100), the authors do not provide any evidence for the convergence of their results with bond dimension $\chi$. While there is no reason to believe the authors did not check this internally it would be beneficial to provide some data showing the convergence with bond dimension.
Additionally, I am wondering why the authors limited their study to only two values of the magnetic field. In this way they verify that the method distinguishes between two dynamical regimes. However, it would be much more interesting to see whether the method can also capture the transition between these regimes in the thermodynamic limit. Furthermore, looking at different values one might be able to understand why "strong" confinement leads to a lack of thermalization while "weak" confinement does not. Is it simply that the case of "weak" confinement is not really confined at all, but only seems to be confined due to short timescales? I believe some of these questions should be explored before the paper can be considered for publication.
Finally, a few minor issues:
- In the abstract the authors mention the Gibbs cannonical ensemble abbreviated as GCE, however, throughout the paper you switch to cannonical Gibbs ensemble CGE. Perhaps this could be made consistent.
- In the main text the authors reference figures 1, 2, 3 and then 11. Since Figure 11 is not part of the main text, perhaps that reference should be replaced with the reference to the relevant Appendix.
- Regarding the Figures, I would recommend reconsidering the use of 3D plots in this case as they simply make it more difficult to see the main results.
- In Figure 3 the Authors claim that there is a significant difference in panels b) and c). I find it difficult to agree with that statement based on what is shown in panels b) and c) alone. Any relevant difference that might be present would only be visible clearly using a log scale.

---

## Round 2 · Referee Report · Anonymous · 2023-3-11

Strengths
1 interesting idea to study thermalization using the full counting statistics
2 suitable and interesting choice of systems, making it possible to study thermalization against dynamical confinement in long-range interacting systems
Weaknesses
1 Unclear, how central quantities (in particular Eq (8), the central quantity studied in the paper) are computed using the matrix-product state techniques applied by the authors
2 Unclear, how valid the numerical results are: MPS+TDVP simulations can lead to qualitatively wrong results for certain initial states, a necessary test is not discussed in the manuscript. Also, the accuracy in particular at long times is determined by the discarded weight, which is not mentioned in the manuscript.
3 English grammar can be improved throughout the manuscript
Report
The manuscript addresses the question for thermalization in long-range ferromagnetic transverse field Ising models using numerical simulations of the time evolution obtained via matrix product states (MPS). The main quantity computed is the full probability distribution function (PDF), which is a very interesting quantity to study for the thermalization dynamics, since it allows one to learn more than by simply looking at the time evolution of a local observable. The results - provided the numerics is not erroneous, see below - are interesting and should be published. However, I feel that SciPost Physics Core would be a more appropriate journal. The following points are important to be addressed by the authors prior to publication:
- it is not clear to me, how the main quantity of their study, the PDF in Eq. (8) is computed using MPS. The reader would benefit from further explaining how this is done using the MPS-approach. If this is too technical, it can be added to the appendix.
- the results look sound; however, from own experience, MPS+TDVP can lead to qualitatively wrong results for certain initial states, in particular product states. Here, the initial state does carry entanglement, which should help the simulations, but the initial bond dimension of 2 is still very small. It is necessary that the authors compare to exact diagonalization results for small systems to make sure the TDVP is not running in a wrong direction for the cases studied. This should be added either in the main discussion of the results, or in the appendix.
- even though not very likely, also finite-temperature results can get stuck or go to wrong results. Again, the initial product state might lead to problems, in particular in connection with the long-range interactions. It is easy to check the accuracy of their results by computing , e.g., the energy as a function of temperature using exact diagonalizations (e.g., using QuSpin) and show a meaningful example in the manuscript (main text or appendix).
- it is not clear, how big their numerical error is. This can be estimated by computing the discarded weight as a function of time (or inverse temperature, resp.), and the authors should provide typical / maximal values of the discarded weight in their simulations. Ideally, comparison to results with other bond dimensions should be shown.
Requested changes
1 provide more details on how calculations were performed, in particular Eq. (8) using MPS (this can be added in the appendix, if too technical).
2 give more insights into accuracy of the numerical results, e.g. by providing the maximum discarded weight encountered in their simulations.
3 benchmark the MPS+TDVP time-evolutions for the given initial state and Hamiltonians for small systems against exact diagonalizations, since TDVP for this initial state might lead to erroneous results (can be included in the appendix).
4 benchmark the finite temperature results against exact diagonalizations (only a generic test for a typical system suffices, can be added to the appendix).

---

## Round 3 · Referee Report · Anonymous · 2023-12-22

Report
The main idea and results of this work are relevant and of interest and I believe the revision did sufficiently address the concerns regarding the novel contribution from this work. Indeed, I have to commend the Authors for the revision of this manuscript. I find that the current version of the manuscript presents the work very clearly with sufficient background and citation to place the work within the broader field. Furthermore, the other points raised in the previous round of Referee reports were also addressed. Nevertheless, a few minor typos and grammatical/stylistic errors remain, but as they are not drastic I leave it to the Authors to decide whether or not they wish to address those before publication (see e.g. multiple definition of LRIM and other typos I will refrain from listing here).
Perhaps I can also take this opportunity to bring up one final suggestion, which would at least to me make things easier to understand quickly. Namely looking at the definition of P(O,t) in equation (5), I feel it would be easier to understand the definition if you simply formulate it in a manner similar to equation (7)
\[P(O,t)=\sum_{\{\sigma_i\}}|C_{\{\sigma_i\}}(t)|^2\delta\left(O-\langle\{\sigma_i\}|\hat O|\{\sigma_i\}\rangle\right)\]
which allows $\{\sigma_i\}$ to run over the complete computational basis.
To keep this short, I think apart from perhaps reformulating the presentation of equation (5) and clearing out the remaining typos and grammatical/stylistic errors, as pointed out above, the manuscript meets the criteria for publication in SciPost Physics.

---

## Round 3 · Author Response

Dear Editor,
We express our sincere thanks to the esteemed referees for dedicating their valuable time to thoroughly reviewing our manuscript and for offering constructive comments. We also extend our apologies for the relative delay in resubmission. In response to the insightful assessments and suggestions provided by the reviewers, we implemented substantial additions and modifications to our previous manuscript. We believe that these enhancements have improved the comprehensiveness of our work, rendering it worthy of publication in SciPost Physics core. Below, we systematically address the questions raised by the referees and the additions we made to the manuscript,
1) Detail on calculating PDF in MPS formalism (Report 2)
We have added Appendix B2 to outline the procedure to calculate full probability distribution function in MPS formalism along with a tensor network diagram in Figure 12.
2) Benchmark TDVP for with exact diagonalization(Report 2)
We have introduced Appendix B3 to address concerns regarding the accuracy of the numerical data presented in the manuscript. In Figure 13, we have bench-marked the TDVP data for 14-site system with exact diagonalization. The details of time evolution with exact diagonalization procedure is presented in Appendix A. In Panel (a) of Figure 13, the absolute error in the energy density is plotted for different system parameters as a function of the inverse temperature. The absolute error remains consistently on the order of 10^(-5) or smaller for all cases. In Panels (b), (c), and (d), we illustrate the absolute error in kink density for various system parameters during the real-time evolution. The errors converge and consistently remain on the order of 10^(-6) for all the cases considered.
3) Evidence on data convergence (Report 1 and 2)
To assess the convergence of TDVP data for larger system sizes, we plot the relative error (the absolute difference between TDVP results with different bond dimensions) in distance to thermalization (DT). This is performed for three values of bond dimensions chi = 60, 90, 128, as shown in Figure 14 in Appendix B3. The errors converge and consistently remain on the order of 10^(-3) or smaller across all cases considered.
4) Brief summary on dynamical regimes (Report 1)
In the first paragraph of section 4, we have added a brief overview of dynamical quantum phase transitions (DQPT) and introduced various dynamical regimes within the long-range Ising model. We have supplemented this information with several references to better understand the background and previous contributions to the field.
5) Choice of post-quench parameters (Report 1)
The choice of two values of transverse fields is based on insights from our previous study (reference number [54] in the References section), which indicated that quenches near the dynamical critical points do not exhibit conclusive equilibration within the time frame feasible with our numerical resources. Consequently, we focused our investigation on two points: h_f = 0.3 and h_f = 0.6. These points reside in the dynamical ferromagnetic and paramagnetic regimes, respectively, and demonstrate a conclusive equilibration.
6) Thermalization and weak confinement (Report 1)
We have introduced Figure 5 to the manuscript, which depicts the post-quench evolution of domain-wall kinks. Because confinement bounds the dynamics of domain-wall kinks during temporal evolution, this parameter serves as a relevant and robust indicator of thermalization. The inclusion of this result further supports the findings shown in Figures 3 and 4.
7) Language and structure (Report 1)
We have edited the manuscript to improve the grammar and language. We relocated the details of the finite-temperature simulations to the appendix. The primary figures have been integrated into the main text. Additionally, for enhanced visualization, we switched panels (a), (b), and (c) of Figures 3 and 4 to the log scale while presenting panels (d), (e), and (f) in 2D for clarity. As a further enhancement, we have introduced Figure 1, encapsulating the central idea of the manuscript.

---

## Round 3 · List of Changes

The list of changes made are as follows;
1) Added appendices A, B2, and B3
2) Added figure 1 in main text
3) Expanded section 4 by adding figure 5
4) re-plotted figure 3 and 4 for better visualization
5) added a paragraph on dynamical regimes on section 4
6) re-located the discussion on simulation of finite temperature states to appendix B1

---

## Editorial Decision

published